# Scaffolding proteins guide the evolution of algal light harvesting antennas

Harry W. Rathbone [1], Katharine A. Michie [1,2], Michael J. Landsberg [3], Beverley R. Green [4] &
Paul M. G. Curmi [1✉]

Photosynthetic organisms have developed diverse antennas composed of chromophorylated proteins to increase photon capture. Cryptophyte algae acquired their photosynthetic organelles (plastids) from a red alga by secondary endosymbiosis. Cryptophytes lost the primary red algal antenna, the red algal phycobilisome, replacing it with a unique antenna composed of αβ protomers, where the β subunit originates from the red algal phycobilisome. The origin of the cryptophyte antenna, particularly the unique α subunit, is unknown. Here we show that the cryptophyte antenna evolved from a complex between a red algal scaffolding protein and phycoerythrin β. Published cryo-EM maps for two red algal phycobilisomes contain clusters of unmodelled density homologous to the cryptophyte-αβ protomer. We modelled these densities, identifying a new family of scaffolding proteins related to red algal phycobilisome linker proteins that possess multiple copies of a cryptophyte-α-like domain. These domains bind to, and stabilise, a conserved hydrophobic surface on phycoerythrin β, which is the same binding site for its primary partner in the red algal phycobilisome, phycoerythrin α. We propose that after endosymbiosis these scaffolding proteins outcompeted the primary binding partner of phycoerythrin β, resulting in the demise of the red algal phycobilisome and emergence of the cryptophyte antenna.

[1] School of Physics, University of New South Wales, Sydney, NSW 2052, Australia. [2] Mark Wainwright Analytical Centre, University of New South Wales, Sydney, NSW 2052, Australia. [3] School of Chemistry and Molecular Biosciences, The University of Queensland, St. Lucia, QLD, Australia. [4] Botany Department, University of British Columbia, Vancouver, BC V6N 3T7, Canada. ✉email: p.curmi@unsw.edu.au

Light harvesting antennas of photosynthetic organisms are incredibly diverse[1–3] and the history of algal endosymbioses presents a challenge in determining their origins[4–8]. Most extant photosynthetic eukaryotes resulted from a single primary endosymbiosis where a cyanobacterium became the ancestral chloroplast. Three eukaryotic lineages arose following this event: red algae, green algae/higher plants, and glaucophytes[9]. Further photosynthetic lineages resulted from secondary endosymbiotic events where heterotrophic protists acquired algal endosymbionts. The modern cryptophyte algae are one such group, with a plastid of red algal origin[4,7]. For cryptophytes, secondary endosymbiosis resulted in a complex set of gene transfers between four loci: the cryptophyte nucleus and the cryptophyte mitochondrion together with the remnant red algal nucleus (nucleomorph) and the plastid of red algal origin[5,8].

Red algae utilise a megadalton-size light harvesting antenna called the red algal phycobilisome (PBS)—comprised of rod structures formed from stacked, hexameric protein-pigment rings emanating from a protein core that is poised above an integral membrane photosystem (Fig. 1a, Supplementary Fig. 1a)[10,11]. Each hexameric ring is built from a conserved heterodimer referred to as the red algal PBS-αβ protomer[12], where red algal PBS α and β subunits are globin fold proteins carrying linear tetrapyrrole chromophores which capture photons[13]. The red algal PBS-αβ protomer represents the minimal unit, where the stability of the two individual α and β subunits is co-dependent[14,15].

The red algal PBS was lost in almost all algal groups with secondary red plastids; the one exception being the cryptophyte algae, which retained one protein subunit from the red algal PBS and acquired a novel protein partner to produce a small, soluble light harvesting protomer. The cryptophyte antenna consists of a dimer of protomers, where each cryptophyte-αβ protomer is composed of a plastid-encoded β subunit that descended from a red algal PBS phycoerythrin β subunit (PE β)[13], and a nuclear-encoded α subunit of previously unknown origin. The cryptophyte α subunit is structurally and evolutionarily distinct from the red algal PBS α subunit. The structure of the cryptophyte-αβ protomer is conserved among cryptophytes[16], while the mature $(α_1β).(α_2β)$ hetero-tetramer exists in two distinct quaternary forms[16–18]. No trace of a cryptophyte α ancestor has previously been identified and the cause of the demise of the red algal PBS antenna is unknown[19].

We present evidence that the cryptophyte α subunit evolved from a previously unidentified family of red algal PBS scaffolding proteins. These scaffolding proteins stabilise PE β subunits that are not part of a conventional red algal PBS-αβ protomer complex. The identifying feature of these scaffolding proteins is a conserved domain (a CALM domain; for cryptophyte α-like motif) that most commonly occurs as repeats. We call the family of scaffolding proteins CaRSPs (CALM repeat scaffolding proteins), where each CALM domain binds a PE β, stabilising and locating it within the red algal PBS. The structure of the CALM: PE β complex is structurally homologous to the cryptophyte-αβ protomer, differing significantly from the red algal PBS-αβ protomer. We have determined the first structures of three CaRSPs by interpreting otherwise unmodelled density in recently deposited cryo-EM maps[10] (Supplementary Table 1). Comparison of cryo-EM maps from two red algal PBS antennas[10,11] shows that the structures of the CaRSPs are conserved across species, as are the positions of their partner PE β subunits within the red algal PBS.

## Results and discussion
### Cryptophyte-αβ homologues in the red algal PBS. Two recent single-particle cryo-EM structures of red algal PBS antennas

(*Porphyridium purpureum*[10] at 2.82 Å resolution and *Griffithsia pacifica*[11] at 3.5 Å resolution) each include 20 modelled PE β subunits that are not part of conventional red algal PBS-αβ protomers (Fig. 1a, Supplementary Fig. 1a). This was surprising, since unpartnered PE β subunits are unstable[14,15]. 16 of these apparently 'lone' PE β subunits (with no PE α partners) are conserved between the two red algal PBS structures (which equates to 8 pairs of PE β subunits in each as both red algal PBS has two-fold symmetry). Of these conserved 'lone' PE β subunits, three pairs are bound to other red algal PBS proteins (Supplementary Note 1). One additional pair of 'lone' PE β subunits in the *P. purpureum* red algal PBS is attached to the C-terminal tail of the red algal PBS rod linker protein $L_R6$[10] potentially providing stability. From the structural models, it is unclear how the remaining 'lone' PE β subunits are stabilised and precisely located at conserved positions.

The C-terminal domain of $L_R6$ that binds PE β in *P. purpureum* forms a flat structure comprised of a β ribbon followed by an α helix (Fig. 1b)[10]. The resulting complex structurally resembles that of the cryptophyte-αβ protomer[16]. Structural alignment of the C-terminal domain of $L_R6$ to published cryptophyte-α subunits produced RMSD values ranging from 0.744 Å over 172 atoms to 1.335 Å over 208 atoms. This provided the first evidence for the existence of a cryptophyte-α-like protein in the red algal PBS (Fig. 1b). The only obvious difference between the two structures is the absence of a chromophore-binding loop in the C-terminal domain of $L_R6$ (Fig. 1b). Examination of the deposited cryo-EM maps (Fig. 1c) confirms the structural similarity seen in the deposited model. Additionally, the map density shows that the N-terminal domain of the PE β subunit (not modelled in the original structure[10]) has adopted a fold that has previously only been seen in cryptophyte-αβ protomers[16–18]. Specifically, when a PE β subunit is part of a red algal PBS-αβ protomer, its N-terminal domain forms a helical hairpin (Fig. 1e, labelled hX and hY), while in cryptophytes, the N-terminal domain is rearranged to form two shorter α helices with a β strand between them (Fig. 1f, labelled hX, hY, and s1) forming a β-sheet with the α subunit β-ribbon (Fig. 1b, c)[18]. Thus, in the red algal PBS, the N-terminal extension of the PE β subunit is metamorphic as it can adopt two folds in the same protein complex (Supplementary Note 2, 3)[20,21] where the fold is templated by the partner[22].

Inspection of the cryo-EM maps in the vicinity of all 4 'lone' PE β subunits unique to each red algal PBS structure plus 10 of the 16 'lone' PE β subunits conserved between the two structures, revealed clear map density interpretable as a stabilising cryptophyte-α-like subunit with the characteristic N-terminal metamorphosis of the PE β subunit (Fig. 1c, d, Supplementary Fig. 1b). Thus, 14 of the 20 'lone' PE β subunits in each red algal PBS form a structure that is homologous to the cryptophyte-αβ protomer by binding to a cryptophyte-α-like domain.

### CALM: Cryptophyte-α-like motif. Structure-based sequence alignment of the refined *P. purpureum* $L_R6$ C-terminal domain against published cryptophyte α subunits shows a similar pattern of sequence conservation compared to alignments without the $L_R6$ C-terminal domain (Fig. 1g). While sequence conservation is low, as is common amongst cryptophyte α subunits, structural conservation is high, as characterised by a low RMSD (above).

Searching sequence databases, we identified 12 other nuclear-encoded red algal proteins potentially related to the C-terminal domain of the $L_R6$ rod linker (Supplementary Figs. 2, 3), including two members of a recently described red algal linker family named "Linker 2"[23]. Multiple sequence alignments and motif searches[24] revealed that most of these proteins contained

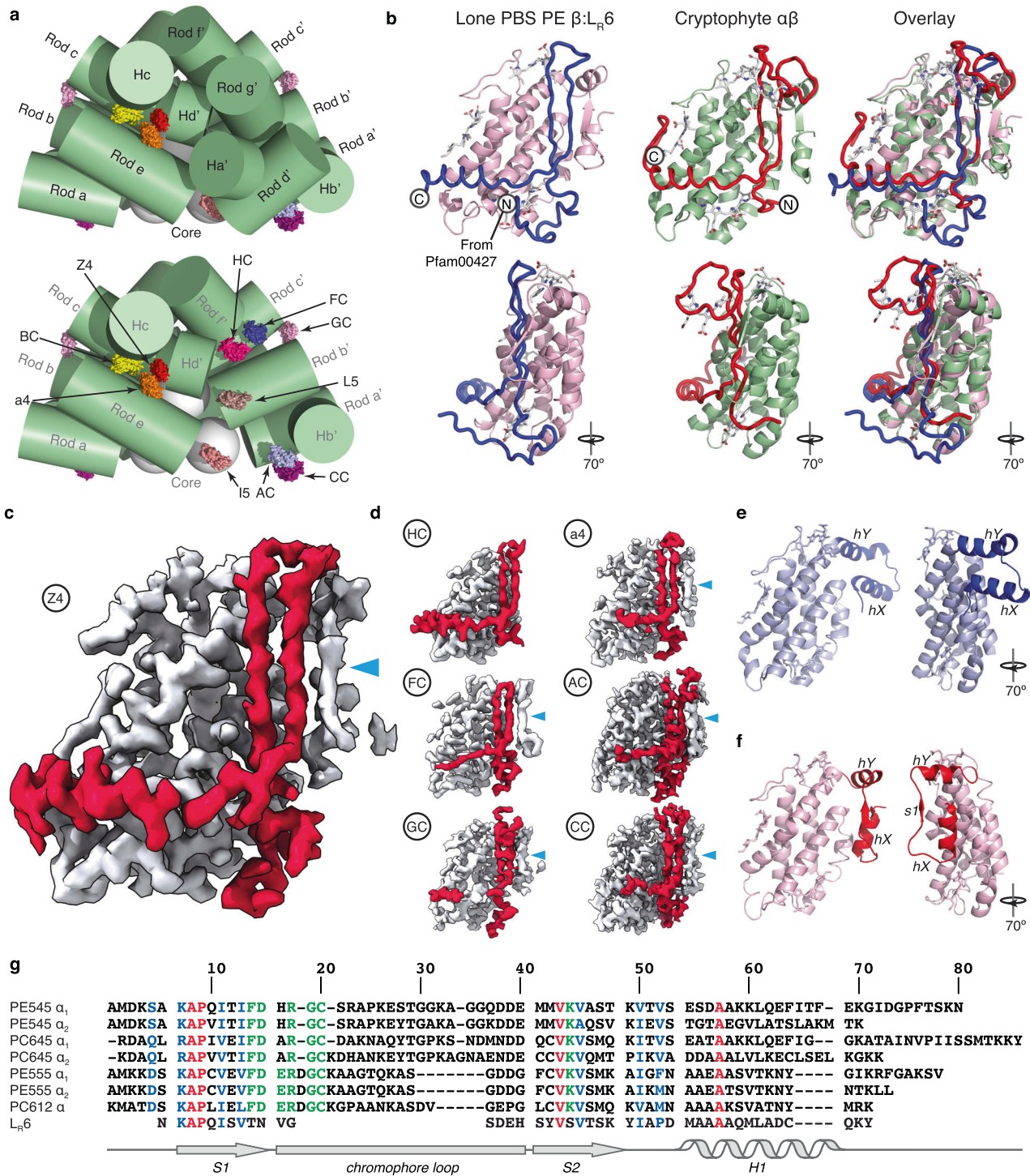

**Fig. 1 CALM domain proteins stabilise 'lone' PE β subunits in the red algal PBS. a** 'lone' PE β subunits (coloured protein surface rendering) peripherally associated to red algal PBS rods (green cylinders) observed in the cryo-EM structure of the red algal PBS from *P. purpureum* (PDB 6KGX)[10]. Lower panel: Rod d', Rod g' and Ha' removed. Labelling is according to Ma et al.[10]. **b** revised model of the C-terminal domain of $L_R6$ with its cognate PE β subunit is structurally homologous to the cryptophyte-αβ protomer. **c, d** sculpted EM map density showing cryptophyte-like protomers from the *P. purpureum* red algal PBS (CALM domains in red and PE β in grey) labelled according to[10]. Cyan triangle indicates the β strand in PE β. **e, f** comparison of metamorphic states of PE β in two rotations with secondary structure labelled: **e** the red algal PBS hexamer form and **f** the cryptophyte-like form. **g** structure-based sequence alignment of cryptophyte α subunits (PDB 1XG0 (PE545), 4LMS (PC645), 4LMX (PE555), and 4LM6 (PC612)) and the C-terminal CALM domain of $L_R6$. Red = identity, blue = conserved, and green = chromophore interacting.

multiple (2–5) copies of a three-part, split motif (Fig. 2d, Supplementary Fig. 3).

The first portion of the motif, which includes an aromatic-rich segment, is not present in cryptophyte α subunits and is not always present in red algal sequences (Supplementary Note 4). The second and third segments of the motif correspond to the β-ribbon plus surrounding residues, which represents the central structural unit of both the C-terminal domain of $L_R6$ and the cryptophyte α subunit

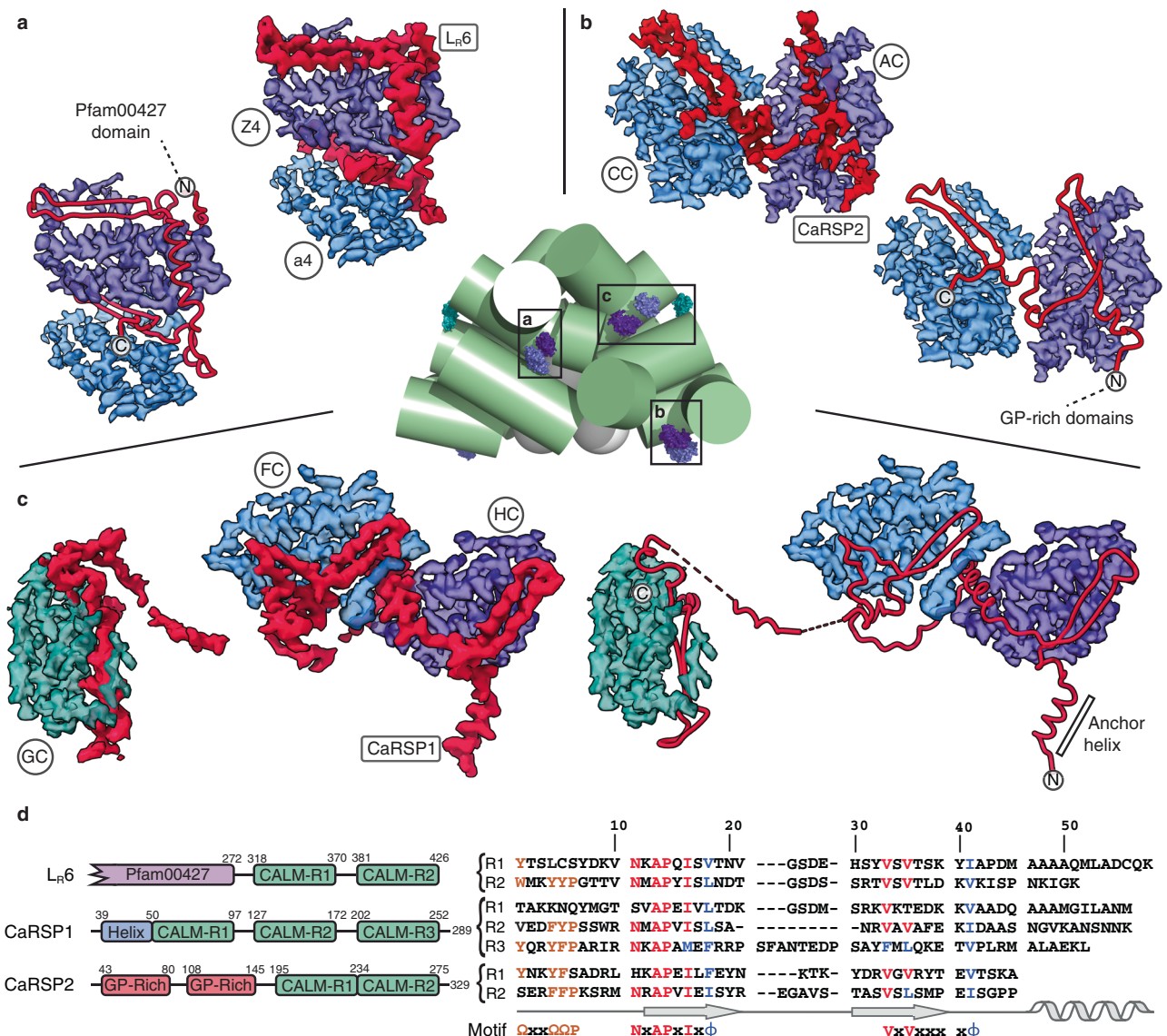

**Fig. 2 The CaRSPs of *P. purpureum*. a–c** sculpted EM map density for L$_R$6, CaRSP2, and CaRSP1, respectively (red), with their associated PE β subunits (coloured purple-blue-teal by their order in the cluster). Both the density and the peptide trace are shown for each CaRSP. A reference to the positions of each cluster is given in the central panel. **d** the domain organization of the three *P. purpureum* CaRSPs (left) and the structure-based alignment of the modelled CALM domains (right). Residue numbers of the beginning and end of each domain are shown (left). Structure and sequence motifs are shown (right; Ω = Y/F/W and ϕ = hydrophobic with orange being the aromatic motif, and red and blue signifying identity and similarity, respectively).

(Fig. 1g, Supplementary Figs. 3, 4a, g, h). The level of sequence identity observed is comparable to that of cryptophyte α subunits (Fig. 1g, Supplementary Fig. 3). We have named the complete motif a CALM domain (cryptophyte α-like motif).

**Scaffolding proteins coordinate lone PE β**. The presence of multiple CALM domains in polypeptide sequences containing them suggests that the proteins may act as scaffolds, capable of stringing together multiple PE β subunits (Fig. 2, Supplementary Fig. 2, Supplementary Note 5). We, therefore, called these scaffolding proteins CaRSPs (CALM repeat scaffolding proteins). Only three CaRSP sequences contain an identifiable globular domain: the red algal PBS linker (Pfam00427) domain in rod linker L$_R$6 (or "Linker 2") (Fig. 2d, Supplementary Fig. 2). Three other CaRSPs contain one or two copies of an unknown 37 residue N-terminal, glycine-proline-rich motif (GP-rich motif; Fig. 2d, Supplementary Fig. 2). The remaining seven CaRSPs

appear to be solely composed of copies of the CALM domain (Fig. 2d, Supplementary Fig. 2).

Further inspection of the deposited maps for the *P. purpureum* red algal PBS revealed evidence for continuous density linking clusters of cryptophyte-like PE β subunits together (Figs. 1a, 2a–c, Supplementary Fig. 5); three such clusters were identified, corresponding perfectly with the three candidate CaRSP sequences that we identified from *P. purpureum*[25]. One of these clusters is L$_R$6, which includes a second CALM domain beyond the original model (Fig. 2a, d)[10]. The other two clusters have three and two scaffolded PE β subunits (Fig. 2b, c) which correspond to the predicted domain architecture of the sequences that we call CaRSP1 and CaRSP2, respectively (Fig. 2d). For the unmodelled region of L$_R$6 and most of the remaining two CaRSPs, we built de novo models including models for the metamorphic N-terminal domain of the associated PE β subunits (Fig. 2a–c, Supplementary Figs. 4, 6, Supplementary Table 1–2, Supplementary Note 6–9). Analysis of each CaRSP structure confirms that they are

scaffolding proteins given that two have an anchoring domain (Fig. 2a, c) and all CaRSPs make multiple contacts with red algal PBS rods as they weave between 'lone' PE β subunits (Supplementary Fig. 4d–f, 5, Supplementary Note 10).

**CaRSPs conserved between red algal PBS antennas**. A comparative analysis of the *G. pacifica* red algal PBS similarly revealed the presence of 14 cryptophyte-like PE β subunits which in this case form two pairs of clusters (Supplementary Fig. 1a); one with three PE β subunits (Supplementary Fig. 1c) and the other with four (Supplementary Fig. 1d). Inspection of the map density reveals the presence of two unmodelled CaRSPs that scaffold each cluster. We built polyalanine chains into the vacant map density as the lack of available sequence data for *G. pacifica* CaRSPs prevented more sophisticated structure interpretation. Remarkably, the two CaRSP:PE β complexes in *G. pacifica* precisely overlay with $L_R6$ and CaRSP1 of *P. purpureum*. The only differences observed in *G. pacifica* were the presence of an additional CALM domain (with PE β subunit) appended to the C-terminus of each cluster (Supplementary Fig. 1, 7) and the absence of N-

terminal anchoring domains seen in *P. purpureum* structures. Thus, CaRSPs are structurally conserved elements of the red algal PBS across red algal species.

**CALM domains stabilise PE β subunits**. The key conserved structural element in both the CALM domain and the cryptophyte α subunits is a β ribbon (and the subsequent extended chain/helix) with a hydrophobic surface on one face (Fig. 1g, Supplementary Fig. 3, 8). The hydrophobic surface of CALM and cryptophyte α (Fig. 3a, b, top) cover almost identical hydrophobic grooves on their respective β subunits (formed by the base of the globin fold and N-terminal metamorphic domain) (Fig. 3a, b, bottom, white-grey groove, outlined in black) with a significant buried surface area for each complex. The binding of either CALM or cryptophyte α to a 'lone' PE β subunit shields the corresponding hydrophobic surfaces from the aqueous medium, stabilising the proteins. We note that the CALM domains, like the cryptophyte α subunits, are likely to fold via a templated mechanism as they adopt flat structures with no independent hydrophobic cores[14,22].

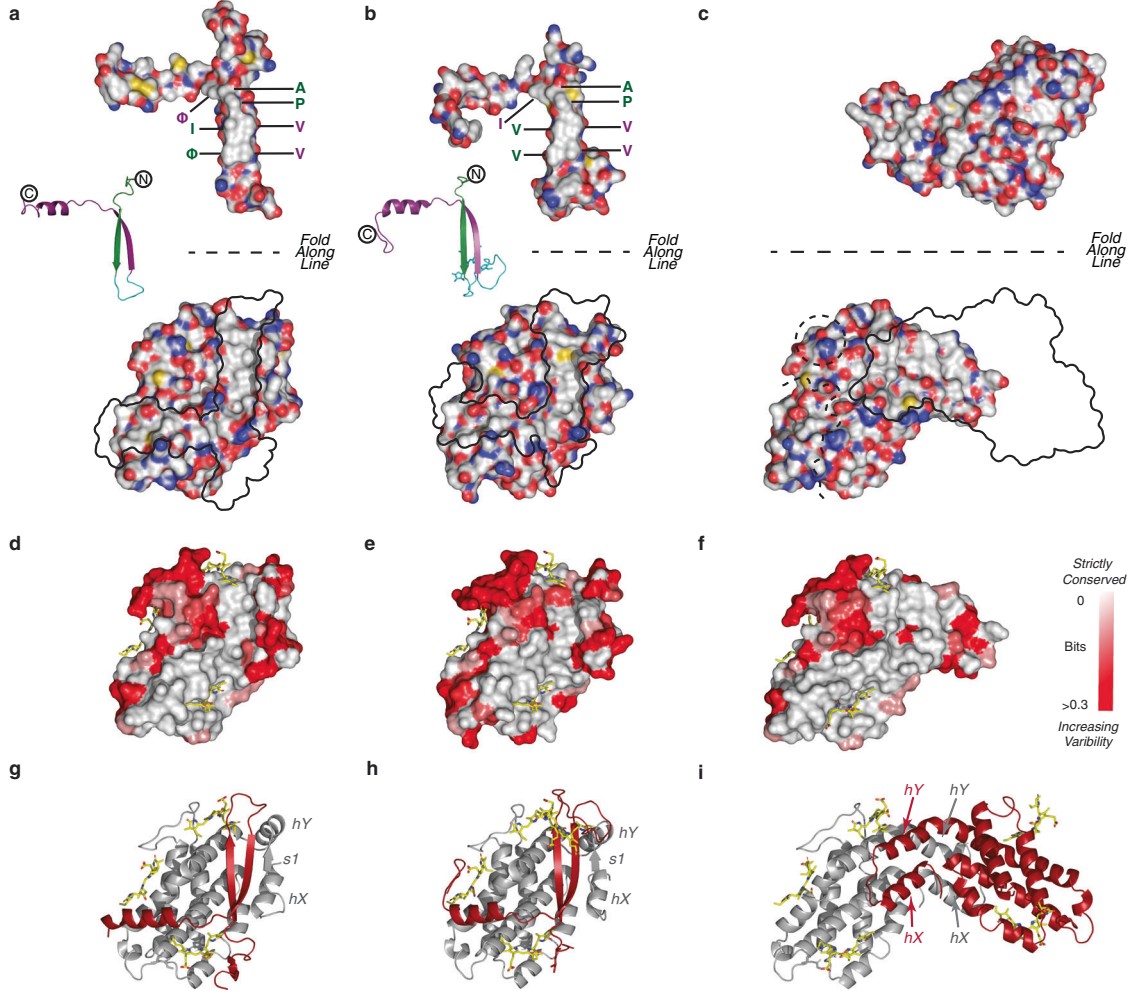

**Fig. 3 The binding surface of PE β is hydrophobic and strictly conserved.** CALM domains (**a** $L_R6$ CALM1), cryptophyte α subunits (**b** PE555 PDB 4LMX), and red algal PBS PE α subunits (**c** PE hexamer PDB 3V57) bind to conserved hydrophobic surfaces of PE β as shown in CPK representation (hydrophobic surfaces appear white). Partner proteins (top panels) are peeled off their respective PE β subunits (lower panels) 180° around the fold line with a black silhouette left in their place. Colour of the residue labels corresponds to the cartoon just to left of each CALM. **d**, **e**, and **f** are mappings of sequence variability (Shannon entropy) onto the molecular surfaces of the PE β structures (white = identity; red = 97% identity or lower; where **d**, **e** and **f** correspond, respectively, to structures in **a**, **b** and **c** above). The conserved and hydrophobic surfaces are congruent. **d** and **f** show the variability for red algal sequences only while **e** includes cryptophyte sequences. **g**, **h**, and **i** are cartoon representations of PE β subunits (grey) with their partner proteins (red).

In the red algal PBS hexamer conformation of PE β, the hydrophobic groove is flattened (Fig. 3c, i). The same residues that are shielded by CALM domains are here shielded by red algal PBS PE α subunits, resulting in a stable red algal PBS-αβ protomer (Fig. 3c, i). The 6 non-CALM bound 'lone' PE β subunits also appear to have found ways to cope with this patch, either through truncation or by co-opting alternative binding partners (Supplementary Fig. 9, Supplementary Note 1). This includes a new linker protein, Linker 3, which we have modelled (Supplementary Fig. 9a–c, Supplementary Note 11).

The PE β residues creating the extended hydrophobic surface in both the red algal PBS and cryptophyte folds are strictly conserved (Fig. 3d–f). The conserved, exposed hydrophobic surface is the reason that isolated red algal PBS β (and red algal PBS α) subunits are unstable and aggregate in vitro[14,15]. The formation of αβ protomers prevents the aggregation and degradation of isolated PE β subunits, which has been observed in cyanobacterial mutants lacking the PBS α subunit gene, ultimately leading to the disruption of the cyanobacterial PBS

architecture[26,27]. The CALM domains perform a similar role in stabilising PE β subunits in both red algae and cryptophytes.

**Evolution of the cryptophyte antenna.** So where did the CALM domain come from? Many red algal PBS linker families (including $L_R6$) evolved and expanded from the cyanobacterial $L_R1$ linker that was transferred to the red algal nucleus following primary endosymbiosis (Fig. 4)[23]. $L_R1$ is composed of two globular domains: the N-terminal Pfam00427 domain and the C-terminal Pfam01383 domain. The latter domain is comprised of a β ribbon, followed by an α helix and another β strand forming a β sheet (Fig. 4 top, inset)[10,28]. The daughter linker families have retained the Pfam00427 domain, anchoring them to a red algal PBS rod. The C-terminal domain diversified, with some comprising a β ribbon, which in $L_R3$ and $L_R6$ is followed by an α helix (Fig. 4, second row insets)[10].

We hypothesise that the CALM domain originated in an early $L_R6$ rod linker, by modification of the ancestral Pfam01383

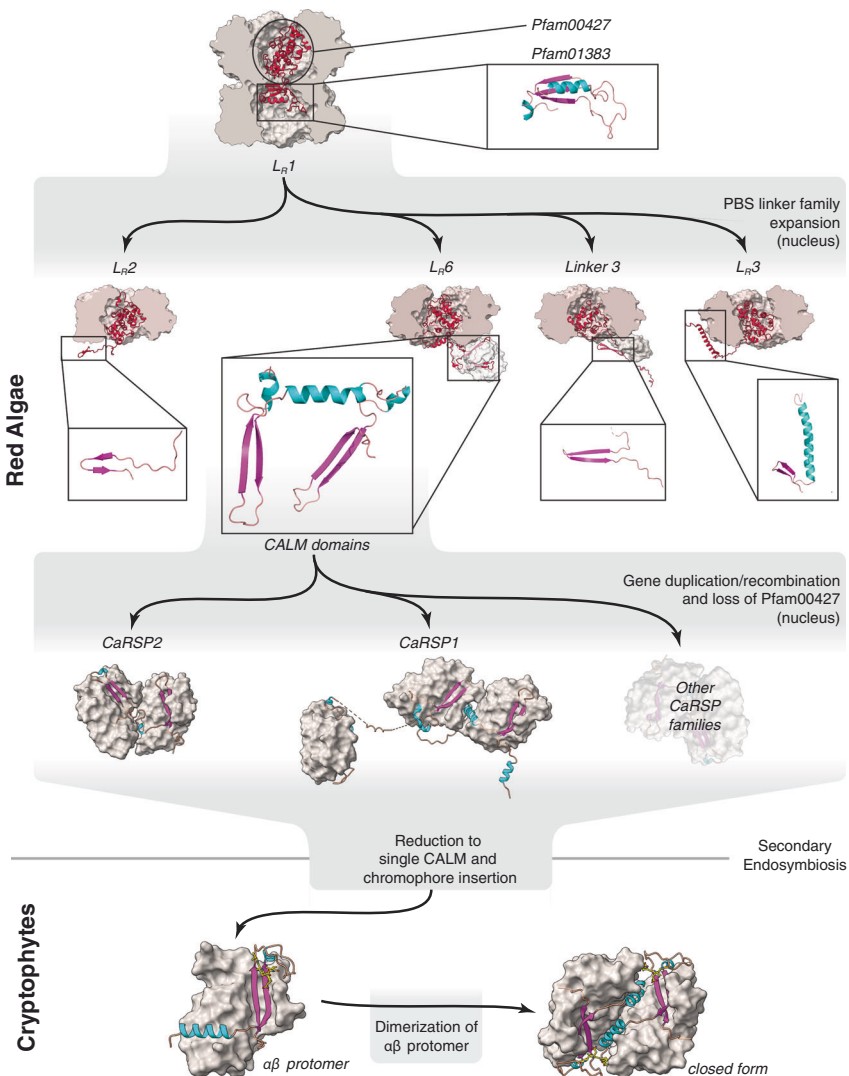

**Fig. 4 Model for the evolution of red algal and cryptophyte antennas.** Top row: Red algae acquire linker protein $L_R1$, which couples red algal PBS ring structures using its Pfam00427 and Pfam01383 domains (inset). Second row: the gene expressing $L_R1$ was transferred to the nucleus, where it expanded and diversified. $L_R2$, "Linker 3", $L_R3$ and $L_R6$ retain C-terminal domains with some similarity to Pfam01383. CALM domains appeared in the red algal PBS linker family ($L_R6$) or, alternatively, in an ancestral CaRSP. Third row: CaRSP proteins lose Pfam00427 domain, expand and diversify in the red algal nucleus. Bottom row: a single CALM protein appeared in the ancestral cryptophyte. It acquired a chromophore plus a plastid targeting sequence. It formed a stable cryptophyte-αβ protomer with a PE β subunit which then dimerised, creating the cryptophyte antenna.

domain, which allows it to bind to the metamorphic PE β (Fig. 4, centre). Gene duplication and recombination likely added additional CALM repeats, resulting in the attachment of more PE β subunits to the red algal PBS, increasing its photon capture cross-section while stabilising the red algal PBS by strapping together rod and hexamer structures. Further rounds of gene duplication and recombination, during which many members of the family lost the Pfam00427 domain, would have given rise to the CaRSP family (Figs. 2 and 4 (third row), Supplementary Fig. 2).

During the secondary endosymbiosis that generated the cryptophytes[29], it is unlikely that all the nuclear-encoded linker genes would have been successfully transferred to the host nucleus at the same time (Supplementary Note 12, 13). This would have lead to instability of the red algal PBS structure. We propose that one of the CaRSPs became the ancestral cryptophyte α subunit. The ability of either CALM domains or red algal PBS PE α subunits to bind to the PE β subunits via the same hydrophobic surface would have facilitated competition between them. It is unclear if phycocyanin β subunits can also bind CALM domains and if they are metamorphic, leading to further competition (Supplementary Note 14). It appears that in the cryptophytes, the primordial cryptophyte α was favoured, leading to the eventual elimination of the PBS PE α subunit and the demise of the red algal PBS, as seen in the mutant *Synechocystis* sp. strain 6803 strain 4 R that is lacking PE α subunit[26].

Even in its primitive state, the cryptophyte αβ protomer would have been able to contribute to light-harvesting, since PE β subunits bind three tetrapyrrole chromophores. Eventually the primordial cryptophyte α subunit would have acquired a chromophore-binding loop with a cysteine for covalent attachment of the fourth chromophore of the cryptophyte-αβ protomer. The ultimate formation of the (αβ)₂ dimer of protomers would have resulted in a stable antenna complex able to capture photon energy and transfer it to the integral membrane photosystems, as seen in extant photosynthetic cryptophytes.

## Methods

**Protein structural alignment**. Structure-based sequence alignment between L$_R$6 and cryptophyte α subunit structures, and between modelled CALM domains presented in this paper was performed by the *super* command in PyMol[30].

Structural similarity between L$_R$6 and cryptophyte α structures (along with the similarity between *P. purpureum* and *G. pacifica* 'lone' PE β subunits) was quantified by RMSD values in PyMol[30] output by the *super* command.

**Sequence analysis**. PsiBLAST was used to find sequences related to the C-terminal domain of the rod linker protein L$_R$6 from *P. purpureum* using the non-redundant NCBI protein database[31]. Initial searches used L$_R$6 residues Ala246 to Lys426 (note: the original cryo-EM structure ends at the equivalent of Tyr371), as using the entire L$_R$6 sequence pulled up proteins containing the red algal PBS linker domain (Pfam00427), which dominated the PSSM matrix. Using the NCBI non-redundant protein database, these searches only pulled up two significant matches, both from red algae: XP_005711070 from *Chondrus crispus* (Irish or carragean moss) and OSX77362 from *Porphyria umbilicalis* (cold water seaweed). The ability of PsiBLAST to find significant hits was increased by limiting the search to red algal sequences only. Iterative PsiBLAST converged on 11 sequences, including L$_R$6 (Supplementary Fig. 2, 3). Initiating PsiBLAST with any of these sequences did not produce new matches. On convergence, the PsiBLAST E-values ranged from 3e-12 to 4e-34.

None of these new sequences contained a red algal PBS linker domain (Pfam00427). However, we noticed that the *P. purpureum* L$_R$6 sequence was identified by Lee et al. 2019[23] as a member of the "Linker 2" clade (POR1447 in their Fig. 4). Of the four other members of this "Linker 2" family, two contain an N-terminal red algal PBS linker domain (Pfam00427) followed by a single CALM domain. Additional BLAST searches, including pattern-hit initiated Phi-BLAST[32] using the most conserved portion of the CALM sequence motif, did not produce any additional members of this family.

GP-rich motifs identified in three sequences (CaRSP2 members) were aligned and then searched for in BLAST. The search only returned one other protein (from *P. purpureum*) that consisted of only two tandem GP-rich motifs.

Sequences were aligned using CLUSTALW[33] as implemented by the NPS@ server[34] or using MUSCLE[35] on the EMBL-EBI webserver[36].

**Identification and alignment of conserved sequence motifs**. Conserved sequence motifs were evident from multiple sequence alignments. To support and generalise motif discovery, we used the MEME web-service[24] from which domains were designated in the sequences. Each repeat of the motif was then extracted and aligned to a reference alignment generated by the structural alignment of L$_R$6 and published cryptophyte α structures using the MAFFT webserver[37,38].

**Model building and refinement of *Porphyridium purpureum* CALM:PE β clusters**. Atomic models were built into vacant densities extracted from the published cryo-EM maps for the phycobilisome (PBS) from the red alga *Porphyridium purpureum*[10]. These maps (downloaded from the Electron Microscopy Data Bank[39]) include the overall map EMD-9976 and the associated local maps EMD-9977 through to EMD-9988.

Initial models for structure determination of cryptophyte-like PE β were made using a chimeric model generated from the globin domain of the crystal structure of the *P. purpureum* B-phycoerythrin ring (PDB 3V57, chain B)[40] and the N-terminal extension (hX-s1-hY) from the cryptophyte *Hemiselmis andersenii* PE555 structure (4LMX, chain B)[16] where 11 residues were mutated so as to match the *P. purpureum* PE β sequence.

These chimeric models were docked to the positions of the cryptophyte-like PE β subunits (PDB 6KGX). PE β cluster AC-CC was reoriented to fit the density of local map EMD-9978. To model the associated CALM domain proteins, three polyalanine chains were then traced through the unmodelled density of the overall map (EMD-9976 for PE β clusters FC-HC-GC and Z4-a4) or one of the local maps (EMD-9978 for PE β cluster AC-CC). Given that the cryo-EM maps for the *P. purpureum* red algal PBS have already been averaged by imposing C2 symmetry[10], we only built models into one of the two asymmetric units of map density.

For the purposes of structure refinement and model building, local maps were produced using customised masks to extract only the density corresponding to each PE β cluster and its corresponding CaRSP. This was accomplished using the phenix.map_box program, which is part of the Phenix suite of programs[41,42]. The amino acid sequence for each CaRSP chain was then fit using the CaRSP proteins identified by the PsiBLAST sequence search. Initial model building was carried out iteratively using the interactive graphics program COOT[43] for manual chain building and phenix.real_space_refine[41,42] for automated real space refinement. ISOLDE[44] within the ChimeraX framework[45] was used preferentially over phenix. refine in the final stages of refinement as the resulting models showed better geometry, with Ramachandran outliers only corresponding to those seen in atomic resolution x-ray structures[17]. Prior to final refinement in ISOLDE, bond length and angle restraints were imposed on the covalent bonds between the phycoerythrobilin (PEB) ligands and the protein during refinements in Phenix. Once the initial models for each of the three clusters were complete, each was taken and treated differently based on the local map resolution or quality.

For the L$_R$6 cluster (Z4-a4), the N-terminal region, corresponding to the Pfam00427 domain (which had already been built into the map[10]) was excluded. Model building and refinement were focused on the C-terminal region of L$_R$6 along with the two PE β subunits. The model was refined in ISOLDE where secondary structure restraints were initially imposed upon helices hX and hY of PE β chains Z4 and a4 and later released. Towards the end of refinement, only select regions (namely around the conserved Ramachandran outlier, Thr75, of each PE β subunit) were refined.

For the CaRSP2 cluster (chains AC-CC), ISODLE was used for refinement, however, given the lower resolution and quality of the map in general, the backbone geometry of each PE β subunit (residues 36–177) was restrained to that of the *P. purpureum* B-phycoerythrin ring crystal structure (3V57 chain D) using ISOLDE's adaptive distance restraints command. Furthermore, secondary structure restraints were imposed on helices hX and hY of the metamorphic N-terminal region of each PE β subunit. The CaRSP2 N-terminal domain (GP-rich region) was not modelled as no density was observed preceding the first CALM domain.

The CaRSP1 cluster (FC-HC-GC) was broken into two segments, due to a large difference in resolution of their respective map regions. The higher resolution segment included PE β chains FC and HC with their associated CALM domains and bridge sequences while the lower resolution segment included PE β chain GC and its associated CALM domain. For the FC and HC segment, refinement was carried out in the same manner to the L$_R$6 cluster while the GC segment was refined in the same manner to the CaRSP2 cluster. Each segment was then recombined in the final model. Some portions of the bridge sequence between CALM domains 2 and 3 were not modelled as the density was too weak to accurately determine the backbone conformation.

Following refinement in ISOLDE and validation in Phenix, small geometry errors (bond length and angle errors) introduced by molecular dynamics in the ISOLDE simulation were resolved in COOT using real space regularisation. In all structures, the determination of resolution for ADP refinement was imperfect. As unfiltered half maps were not deposited in the EMDB, we could not estimate the resolution by Fourier shell correlation for each cluster, and resolution was instead estimated by visual inspection of Supplementary Fig. 2 from Ma et al.[10]. The CaRSP1 cluster had an additional challenge as it was refined (as stated above) in

two segments with different resolutions supplied for refinement. When the two segments were brought back together, the average resolution was quoted as the global resolution of this cluster for calculation of B-factors. As such, B-factors should be regarded as a guide to local model quality. Correlation coefficients between map and model (CC-values) were calculated for each residue to give a measure of local map/model quality. These are mapped onto each structure for comparison between proteins and regions within each protein (Supplementary Fig. 6).

Model validation statistics for deposited structures can be found in Supplementary Table 1. PDB accession numbers can be found in Supplementary Table 1.

**Model building and refinement of other *P. purpureum* proteins**. In addition to extending the model of L<sub>R</sub>6 to include the PE β-binding CALM domain and building models for the novel red algal PBS proteins CaRSP1 and CaRSP2, we were also able to build the structure for an additional linker protein. Using the sequence for the C-terminal domain of "Linker 3", a protein previously identified by Lee et al.[23] but which is not modelled in the Ma et al. structure[10], it was possible to build a complete model into the vacant map density sitting above 'lone' PE β chain BC (as per PDB 6KGX; Supplementary Fig. 9a–c). The chain of Linker 3 was built using COOT into this stretch of clear map density and was then refined using ISOLDE. Building a complete structure was not possible; the stretch between the N-terminal Pfam00427 domain and the β hairpin did not have clear enough density to allow unambiguous model building, likewise the chain after the β hairpin. One small section was however modelled only for use in Supplementary Fig. 4f highlighting a two-residue motif. We also note that there is a large disparity between the map quality of PE β chain BC and Linker 3 potentially due to low occupancy of PE β chain BC. The underlying PE β chain was fit by taking chain D of PDB 3V57, truncating the structure to remove helices hX and hY and then rigid body fitting into the density in COOT. As such, the B-factors for the PE β subunits are directly inherited from chain D of PDB 3V57. Some rotamers along the interface between PE β and Linker 3 were changed to accommodate the bound linker in COOT as well.

Finally, we also refined the structure of L<sub>R</sub>2, whose C-terminal β ribbon was quickly simulated in ISOLDE to refine the secondary structure, which was not apparent in the deposited structure. Beyond clarifying the secondary structure, this refinement did not generate any large changes to the overall structure.

Model validation statistics for deposited structures can be found in Supplementary Table 1. PDB accession numbers can be found in Supplementary Table 1.

**Model building and refinement of *Griffithsia pacifica* CALM:PE β clusters**. The conservation of the CaRSP proteins from *P. purpureum* in the *G. pacifica* red algal PBS was clear from visual inspection of the two cryo-EM maps. However, the absence of pre-existing sequence data for CaRSP proteins from the latter, combined with the lower quality of the *G. pacifica* map[11] meant that it was impossible to build these structures de novo. Instead, using an analogous, but albeit more limited approach to what was used to determine structures of the corresponding structures from *P. purpureum*, we produced structures of putative CaRSP1-like and L<sub>R</sub>6-like proteins from *G. pacifica* that reveal the Cα backbone trace of these proteins, including the presence of CALM motifs that establish interactions with the corresponding PE β subunits.

Maps of the *G. pacifica* red algal PBS were retrieved from the Electron Microscopy Data Bank[39], including the overall map EMD-6769 and the associated local maps EMD-6758 to EMD-6768. Structures were built initially by taking the *P. purpureum* models (PE β subunit plus CALM domain) and mutating the side chains of the PE β subunits to match the corresponding sequences in *G. pacifica*. These improved models for CALM:PE β complexes were then docked into the map density occupied by the 'lone' PE β subunits in the originally deposited structure of the *G. pacifica* red algal PBS (PDB 5Y6P). To complete the structures of the CaRSPs, using the overall map only (EMD-6769), we then built polyalanine chains into the interconnecting density between the CALM domains within the two PE β clusters (jJ-jL-jP-jM and kX-kW-kV). Polyalanine chains were manually built into the overall map in COOT[43] and subjected to a minimal structural refinement using phenix.real_space_refine[41,42]. We note that these models were only subsequently used to render map density for the CALM domains and to compare the CaRSP structures from the two red algal PBSs.

**Conserved surface calculations**. All complete red algal PE β subunit sequences (not including fragments and partial sequences) were drawn from UniProtKB totalling 83 sequences (plus a further 10 complete cryptophyte sequences). The sequence variability was quantified by the Shannon entropy[46] of each residue site over the 83 sequences (or 93 sequences in the case of red algae and cryptophytes together). This was calculated in the numerical computation program *Mathematica* using the standard Shannon entropy formula (in units of bits; Eq. 1),

$$H = -\sum_{i=\mathrm{Ala}}^{\mathrm{Tyr}} \frac{n_i}{N}\log_2\frac{n_i}{N}$$

**Equation 1**. Entropy calculation per peptide site

where the summation is over the 20 naturally occurring amino acids (from Ala to Tyr). N is the total number of sequences and $n_i$ is the number of sequences with a particular amino acid for that site. The entropy (red algae only) corresponding to each residue site in the sequence was mapped onto the 3D structures of the red algal PBS ring form (using 3V57 chain B; Fig. 3f) and the cryptophyte-like 'lone' PE β conformation (using chain a4 of the L<sub>R</sub>6 cluster; Fig. 3d). The entropy (red algae and cryptophytes) corresponding to each residue was also mapped onto the cryptophyte PE β PE555 (chain B of PBD: 4LMX; Fig. 3e). Molecular surfaces were rendered in PyMol. Entropy was mapped onto protein surfaces by replacing B-factors with entropies in the corresponding PDB files. Surfaces were then coloured by B-factor and colour scale was capped at H < 0.3 to identify strictly conserved residues. For reference: a Shannon entropy of zero implies that a particular residue is completely conserved; the maximum Shannon entropy is ~4.32 (= $\log_2 20$) and this implies uniform probability for any amino acid at this site (no conservation); finally, a Shannon entropy of 0.3 (as in Fig. 3d–f) corresponds to approximately 97% sequence identity (with all other residues at equal probability). The addition of the cryptophyte sequences produced minimal change to the Shannon entropy surface apart from a slightly higher variability for a few residues (compare Fig. 3e–d). The surface formed by the metamorphic region is highly conserved (Fig. 3f). Mapping the conserved residues of red algal PE β subunits onto the red algal PBS ring and cryptophyte-like conformations attests to the highly conserved nature of the hydrophobic surface upon which various partner proteins bind (Fig. 3f and 3d, respectively).

**Rendering of structural data**. EM map density surfaces (Fig. 1c, d, 2a–c, Supplementary Fig. 1b–d) and EM map density meshes (Supplementary Fig. 4, 9a–c) were rendered in *ChimeraX*[45]. All other protein structures, correlation coefficient plots, protein surfaces (those coloured by atom and by sequence entropy), and red algal PBS rod diagrams were rendered in *PyMol*[30].

**Reporting summary**. Further information on research design is available in the Nature Research Reporting Summary linked to this article.

## Data availability

Atomic coordinates for models generated in this work have been deposited with the Protein Data Bank with accession codes for CaRSP1 plus associated PE beta subunits as 7LIX; CaRSP2 plus associated PE beta subunits as 7LIY; LR6 plus associated PE beta subunits as 7LIZ; and Linker 3 plus associated PE beta subunit as 7LJ0. Cryo-EM maps used in this work were downloaded from the Electron Microscopy Databank with accession codes EMD-9976 through to EMD-9988 for *Porphyridium purpureum* (with associated model PDB 6KGX) and EMD-6758 through to EMD-6769 for *Griffithsia pacifica* (with associated model PDB 5Y6P). Models were built using publicly available structural models: PDB 4LMX and PDB 3V57. Accession codes for protein sequences used in this work are KAA8497087 for Linker 3 and KAA8491180, KAA8495560, KAA8491883, XP_005716950, XP_005711070, PXF45458, PXF49306, OSX75119, OSX70945, OSX69271, OSX70368, OSX77362, OSX68985 for CaRSPs, as shown in Supplementary Fig. 2. Publicly available protein structures used for structure-based sequence alignment (Fig. 1) were PDB: 1XF6, 4LMS, 4LMX, 4LM6. All other data and materials are available on reasonable request. Requests for materials should be directed to the corresponding author.

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

## Acknowledgements

We thank Andrew Torda for advice regarding protein motifs and Tristan Croll for parameterizing chromophores for use in ISOLDE. We acknowledge the use of the facilities in the Structural Biology Facility within the Mark Wainwright Analytical Centre —UNSW, funded in part by the Australian Research Council Linkage Infrastructure, Equipment and Facilities Grant: ARC LIEF LE190100165. This research was supported by Australian Research Council Discovery Grants (DP180103964) and U.S. Air Force Office of Scientific Research through the Asian Office of Aerospace Research and Development (FA2386-17-1-4101) grants to PMGC. H.W.R. is supported by a scholarship from the Australian Government Research Training Program.

## Author contributions

This research was conceptualised by P.M.G.C. and H.W.R. Building and refinement of cryo-EM structures was carried out by H.W.R., P.M.G.C., and M.J.L. Sequence analysis was carried out by H.W.R., B.R.G., and P.M.G.C. All authors contributed to the writing of the manuscript.

## Competing interests

Authors declare no competing interests.
