## [Peer Review File · Nature Communications]

REVIEWERS' COMMENTS

Reviewer #1 (Remarks to the Author):

One of the most interesting questions in molecular evolution is where and how new genes arise. This is not a question that is easy to answer due to the often rapid evolution of sequences following gene duplication. Furthermore, interest in understanding the diverse way in which autotrophs form and utilize their light harvesting apparatuses have many important applications from biofuel to food production. Rathbone et al. provide compelling evidence for a hypothesis on the origin of the novel α subunit in cryptophyte phycobiliproteins. Interestingly, the genes that encode this subunit are found in the nuclear genome that was inherited from the heterotroph eukaryote and travel across 5 membranes to form the cryptophyte phycobiliprotein. These characteristics makes this an excellent study system for understanding the evolution of novel genes and photosynthesis. Below I provide several suggestions to help the authors clarify and improve their manuscript.

1. I think it may be easier to follow along if the authors consistently used "red algal" before PBS when referring to the PBS of red algae.
2. Why is the term "cryptophyte- $\alpha\beta$ protomer" used and not the more common term "cryptophyte phycobiliprotein"?
3. I think the nuclear location of the α subunit in cryptophytes should be clearly stated at the beginning of the manuscript. Furthermore, it should be clarified that the nucleomorph genome of cryptophytes has been shown to be the ancestral red algal nuclear genome and not the cryptophyte nuclear genome. This was not clearly stated in the manuscript and will surely cause confusion among people who don't know the four genome system of cryptophytes.
4. I think the authors should make use of the ~ 20 α subunit genes of *Guillardia theta* (see also Kieselback et al. 2018) and several other published amino acid sequences and attempt to reconstruct the ancestral amino acid sequence of the cryptophyte α subunit and compare that to the Lr6 amino acid sequence. Also, I may have missed this, but what are the NCBI accession numbers for the cryptophyte α subunit sequences used in Fig. 1?

Reviewer #2 (Remarks to the Author):

This is an impressive paper that analyzes the unique phycobiliprotein antenna complexes found in the cryptophyte algae. The cryptophyte complexes have long been mysterious, as they are composed of a beta subunit similar to that found in cyanobacterial and red algal phycoerythrins, plus an alpha subunit of cryptic origin. The current paper analyzes the two recent cryo-EM structures of red algae (the evolutionary precursor to the cryptophytes) and identifies a linker protein that is similar in structure to the cryptophyte alpha protein. The authors have done an extensive analysis of all these proteins and have concluded that the linker protein is the evolutionary origin of the cryptophyte alpha protein. I found the analysis to be convincing. The manuscript contains an extensive amount of supplementary data and even a Supplementary Discussion, which I have never seen before in a published paper. I suspect that much of this material has originated in a thesis. My suggestion is to eliminate this material from the current manuscript and possibly publish it elsewhere as a follow-up to this paper.

Reviewer #3 (Remarks to the Author):

Curmi and co-workers describe a very novel idea that is aimed to provide an explanation on the appearance of the alpha-subunit of the cryptophyte photosynthetic antenna. This complex is a derivative of the widespread Phycobilisome antenna found in cyanobacteria and red-algae. The

cryptophytes, a eukaryotic organism that is rather unique in the evolutionary process that led to this antenna "losing" the gene encoding for the typical phycobiliprotein alpha-subunit of phycoerythrin (PE), along with the PC and APC components. However, it appeared to "gain" a new gene, encoding for completely different a subunit, with a single chromophore and stabilizing the beta-PE subunit. The appearance of this novel gene could not be easily reconciled by simple screening of homologues in the gene banks. The authors carefully looked at the new cryo-EM structures of the entire PBS from two red-algae that had identified β -PE subunits that were not associated within regular PE trimers. Utilizing the EM data deposited in the data bank, they used the calculated electron density maps to identify unoccupied electron density in close proximity to these "lone" beta-PE subunits. They hypothesized that these densities might be occupied by stabilizing proteins that evolved later into the cryptophyte alpha-PE (CaRSPs). Since the structure of the cryptophyte alpha/beta-PE subunits are known, they were able to build the CaRSPs into these empty densities, as homologues of the cryptophyte alpha-PE subunits, essentially proving their hypothesis, and then analyzing in great detail the associations between these subunits within the PBS. In the process, they improved the structure of the LR6 protein and a novel linker 3 protein. The resulting protein structures, based on reinterpretation of the cryo-EM data, have been deposited in the PDB (no accession numbers provided).

The manuscript is excellent, and I highly recommend that Nature Communications accept it for publication.

I have one small comment, that the authors can entertain, but should not delay acceptance. The authors added a supplementary discussion section, which is very extensive, and thus to the most part makes sense in removing from the main text. Section N however is very interesting as it relates to a cardinal question in all phycobiliprotein assembly: how is it that the correct assemblies always occur, even though the homologies are high enough to support at least some competition. There are a few papers that addressed this issue (for instance on the assembly of the different APC forms in the cores of the PBS, similar to that shown in Fig. 3), but as the authors themselves state – it could be that minor heterogeneities in assembly are just not easily seen against the background of the normal assemblies. This has also recently been seen in crystal structures of PBP proteins that are heterogeneous in assembly. My point here is that this section (at the very end) might be missed by readers. Perhaps a greater part might be transferred to the main text in some fashion?

REVIEWERS' COMMENTS

Reviewer #1 (Remarks to the Author):

One of the most interesting questions in molecular evolution is where and how new genes arise. This is not a question that is easy to answer due to the often rapid evolution of sequences following gene duplication. Furthermore, interest in understanding the diverse way in which autotrophs form and utilize their light harvesting apparatuses have many important applications from biofuel to food production. Rathbone et al. provide compelling evidence for a hypothesis on the origin of the novel α subunit in cryptophyte phycobiliproteins. Interestingly, the genes that encode this subunit are found in the nuclear genome that was inherited from the heterotroph eukaryote and travel across 5 membranes to form the cryptophyte phycobiliprotein. These characteristics makes this an excellent study system for understanding the evolution of novel genes and photosynthesis. Below I provide several suggestions to help the authors clarify and improve their manuscript.

1. I think it may be easier to follow along if the authors consistently used "red algal" before PBS when referring to the PBS of red algae.

As requested, we have consistently used "red algal" before PBS when referring to the PBS of red algae.

2. Why is the term "cryptophyte- $\alpha\beta$ protomer" used and not the more common term "cryptophyte phycobiliprotein"?

The term "cryptophyte- $\alpha\beta$ protomer" specifically refers to the minimal structural unit that combines with a second copy to form a "cryptophyte phycobiliprotein" where the latter is a dimer of protomers. These terms cannot be interchanged. By using this terminology, we remove ambiguity caused by terms such as dimer vs. tetramer vs. dimer of dimers – which have previously been used to describe either the cryptophyte- $\alpha\beta$ protomer or the cryptophyte phycobiliprotein.

3. I think the nuclear location of the α subunit in cryptophytes should be clearly stated at the beginning of the manuscript. Furthermore, it should be clarified that the nucleomorph genome of cryptophytes has been shown to be the ancestral red algal nuclear genome and not the cryptophyte nuclear genome. This was not clearly stated in the manuscript and will surely cause confusion among people who don't know the four genome system of cryptophytes.

We have added the following sentence to the end of the first paragraph in the Introduction:

"For cryptophytes, endosymbiosis resulted in a complex set of gene transfers between four loci: the cryptophyte nucleus and the cryptophyte mitochondrion together with the remnant red algal nucleus (nucleomorph) and the plastid of red algal origin (refs 5, 8)"

4. I think the authors should make use of the ~20 α subunit genes of *Guillardia theta* (see also Kieselback et al. 2018) and several other published amino acid sequences and attempt to reconstruct the ancestral amino acid sequence of the cryptophyte α subunit and compare that to the Lr6 amino acid sequence. Also, I may have missed this, but what are the NCBI accession numbers for the cryptophyte α subunit sequences used in Fig. 1?

As per the letter from the Editor (Dr Pattison), this suggestion is "considered ... optional". We agree with the Editor, in that ancestral reconstruction represents a separate piece of work, which we choose not to pursue it here. It would be complicated by the poor sequence conservation in cryptophyte alpha subunits and the complex genetic history of their nuclear genomes.

Regarding the sequences used in Fig. 1: we have added the PDB accession codes in the figure caption, as the PDB is the source of the sequences that we have used, additionally this is a structure-based sequence alignment hence the 3-D structures are paramount.

Reviewer #2 (Remarks to the Author):

This is an impressive paper that analyzes the unique phycobiliprotein antenna complexes found in the cryptophyte algae. The cryptophyte complexes have long been mysterious, as they are composed of a beta

subunit similar to that found in cyanobacterial and red algal phycoerythrins, plus an alpha subunit of cryptic origin. The current paper analyzes the two recent cryo-EM structures of red algae (the evolutionary precursor to the cryptophytes) and identifies a linker protein that is similar in structure to the cryptophyte alpha protein. The authors have done an extensive analysis of all these proteins and have concluded that the linker protein is the evolutionary origin of the cryptophyte alpha protein. I found the analysis to be convincing. The manuscript contains an extensive amount of supplementary data and even a Supplementary Discussion, which I have never seen before in a published paper. I suspect that much of this material has originated in a thesis. My suggestion is to eliminate this material from the current manuscript and possibly publish it elsewhere as a follow-up to this paper.

We have renamed the Supplementary Discussion to Supplementary Notes in accordance with Nature Communications style. We have kept this material as Supplementary Notes, as it is integral to supporting the main manuscript.

Reviewer #3 (Remarks to the Author):

Curmi and co-workers describe a very novel idea that is aimed to provide an explanation on the appearance of the alpha-subunit of the cryptophyte photosynthetic antenna. This complex is a derivative of the widespread Phycobilisome antenna found in cyanobacteria and red-algae. The cryptophytes, a eukaryotic organism that is rather unique in the evolutionary process that led to this antenna “losing” the gene encoding for the typical phycobiliprotein alpha-subunit of phycoerythrin (PE), along with the PC and APC components. However, it appeared to “gain” a new gene, encoding for completely different a subunit, with a single chromophore and stabilizing the beta-PE subunit. The appearance of this novel gene could not be easily reconciled by simple screening of homologues in the gene banks. The authors carefully looked at the new cryo-EM structures of the entire PBS from two red-algae that had identified β -PE subunits that were not associated within regular PE trimers.

Utilizing the EM data deposited in the data bank, they used the calculated electron density maps to identify unoccupied electron density in close proximity to these “lone” beta-PE subunits. They hypothesized that these densities might be occupied by stabilizing proteins that evolved later into the cryptophyte alpha-PE (CaRSPs). Since the structure of the cryptophyte alpha/beta-PE subunits are known, they were able to build the CaRSPs into these empty densities, as homologues of the cryptophyte alpha-PE subunits, essentially proving their hypothesis, and then analyzing in great detail the associations between these subunits within the PBS. In the process, they improved the structure of the LR6 protein and a novel linker 3 protein. The resulting protein structures, based on reinterpretation of the cryo-EM data, have been deposited in the PDB (no accession numbers provided).

The manuscript is excellent, and I highly recommend that Nature Communications accept it for publication. I have one small comment, that the authors can entertain, but should not delay acceptance. The authors added a supplementary discussion section, which is very extensive, and thus to the most part makes sense in removing from the main text. Section N however is very interesting as it relates to a cardinal question in all phycobiliprotein assembly: how is it that the correct assemblies always occur, even though the homologies are high enough to support at least some competition. There are a few papers that addressed this issue (for instance on the assembly of the different APC forms in the cores of the PBS, similar to that shown in Fig. 3), but as the authors themselves state – it could be that minor heterogeneities in assembly are just not easily seen against the background of the normal assemblies. This has also recently been seen in crystal structures of PBP proteins that are heterogeneous in assembly. My point here is that this section (at the very end) might be missed by readers. Perhaps a greater part might be transferred to the main text in some fashion?

We agree with the reviewer regarding the importance of Section N in our original submission (corresponds to Supplementary Note 14 in the revised manuscript). To prevent the reader missing this discussion, we have added the following sentence to the penultimate paragraph of the main text:

“It is unclear if phycocyanin β subunits can also bind CALM domains and if they are metamorphic, leading to further competition (Supplementary Note 14).” (Page 10, lines 3-5)